# Real-Time Identification Algorithm of Daylight Space Debris Laser Ranging Data Based on Observation Data Distribution Model

**DOI:** 10.3390/s25072281

**Published:** 2025-04-03

**Authors:** Yang Liu, Xue Dong, Jian Gao, Bowen Guan, Yanning Zheng, Zhipeng Liang, Xingwei Han, He Dong

**Affiliations:** 1Changchun Observatory, National Astronomical Observatories, Chinese Academy of Sciences, Changchun 130117, China; liuyang@cho.ac.cn (Y.L.); gaoj@cho.ac.cn (J.G.); zhengyn@cho.ac.cn (Y.Z.); liangzp@cho.ac.cn (Z.L.); hanxw@cho.ac.cn (X.H.); 2University of Chinese Academy of Sciences, Beijing 100049, China; 3College of Information Technology, Jilin Normal University, Siping 136000, China

**Keywords:** daylight debris laser ranging, real-time data identification, chi-square test

## Abstract

**Highlights:**

**What are the main findings?**
A statistical distribution-based algorithm is proposed to distinguish weak echo signals from intense daylight background noise, achieving real-time identification of space debris laser ranging data within 1 s.The method successfully detects echo signals with intensities as low as 0.09 photons per pulse under high-noise conditions (background noise rate: 2 × 10^7^ photons/s), surpassing the traditional intensity threshold constraints.

**What is the implication of the main finding?**
Enables continuous daylight tracking and precise orbit determination of space debris in low signal-to-noise ratio (SNR) environments, which is critical for spacecraft safety.Leverages statistical distribution disparities instead of signal intensity, offering a universal framework for weak signal extraction in photon-starved regimes.

**Abstract:**

In an effort to accomplish the real-time acquisition of the laser ranging results of space debris during the daylight and enhance the observation success rate, this paper establishes a joint distribution model of noise and echo signals grounded on the intensity distribution law of the daylight background noise. Through an in-depth analysis of the measurement characteristics of single-photon detectors, a real-time recognition algorithm based on the disparity in statistical distribution is put forward. This algorithm partitions the observation data into intervals of equal length. It then employs the goodness-of-fit test of the geometric distribution to ascertain the data distribution law. Subsequently, it locates the interval in which the echo signal resides by analyzing the contribution degree of the chi-square statistic. The experimental outcomes indicate that under the circumstances of a laser frequency of 400 Hz and a background noise photon rate of 2 × 10^7^ photons per second, the algorithm is capable of achieving real-time recognition of the echo interval for an intensity of 0.09 echo photons per single pulse within 1 s. This breakthrough resolves the critical challenge of daylight echo discrimination in high-noise environments. This method overcomes the constraints of the traditional signal intensity threshold and offers a novel technical approach for the tracking and precise orbit determination of space debris in a low signal-to-noise ratio environment.

## 1. Introduction

With the increasing frequency of space activities, the amount of space debris has surged dramatically, posing severe threats to the safety of in-orbit spacecraft [1,2]. Satellite laser ranging (SLR) technology, renowned for its high single-point precision, has been widely applied in the precise measurement and orbit prediction of space debris [3,4,5,6]. However, constrained by the accuracy of debris orbit prediction, current laser ranging can only be conducted during the observable morning and evening twilight periods and the daytime, with the daytime observable window accounting for over two-thirds of the total [7]. As its diffuse reflection echo signals exhibit dispersed energy and weak intensity, research on the weak signal identification in space debris laser ranging holds significant application value and practical importance for achieving continuous, stable, and automated tracking observations of space debris laser ranging, as well as rapid precise orbit determination and prediction of large-scale space debris [8,9,10].

Addressing the complex challenge of low signal-to-noise ratio (SNR) echo identification, the existing studies primarily fall into three categories. The first involves temporal correlation analysis. For example, Ref. [11] employs residual comparison to filter valid signals, yet its detection rate remains unsatisfactory for high-orbit targets. The second category comprises statistical filtering methods. Works like Refs. [12,13] design robust estimation windows based on the Poisson distribution characteristics, but these approaches rely on the assumption that the signal density surpasses the noise. The third category focuses on photon accumulation strategies. Refs. [14,15,16] propose time-correlated photon accumulation methods; however, in sparse-signal scenarios, the false detection rates escalate significantly. All of these methods require the echo density to exceed the noise density entirely. In practice, due to the detector characteristics in laser ranging, the noise density at the moment of range gate activation often surpasses the signal density at echo arrival, leading to the frequent misidentification of noise as valid signals and consequent algorithm failure.

To tackle these issues, the current research concentrates on echo signal identification under low-SNR conditions. The traditional manual interpretation methods, though adaptable to weak signals, suffer from inefficiency and reliance on empirical expertise. While the existing approaches have improved the detection efficiency to some extent, their performance remains constrained by the assumption that the signal intensity must exceed the noise density. In daytime observations, where the noise density frequently dominates the signal density, the current algorithms exhibit sharply increased false detection rates, necessitating manual post-processing. To address this, this paper abandons conventional intensity thresholding and proposes a real-time identification algorithm based on the statistical distribution characteristics of the noise and echoes. By establishing a joint distribution model of noise and echo signals, combined with dynamic interval partitioning and goodness-of-fit testing, we achieve signal identification under strong noise interference, providing theoretical support and technological breakthroughs for real-time ranging in complex noise environments.

## 2. Establishment and Analysis of Observation Data Distribution Model

Based on the working principles of single-photon detectors, each measurement of an effective signal constitutes an independent event, while in a single measurement, the reception of noise signals and echo signals are mutually exclusive events. Although the detector cannot distinguish between noise and echo signals during the measurement process, it can be assumed that the intensity of the received background noise and echo signals remains constant within a specific time window. Consequently, a distribution model for the measurement data can be established according to the system’s operational principles. By analyzing the goodness-of-fit between the measurement data and the model, the information contained in the measurement data can be effectively extracted.

### 2.1. Sky Background Noise and System Model

Sky background noise is the primary interference source in daytime satellite laser ranging. Sunlight undergoes scattering by atmospheric molecules and aerosol particles, as well as thermal radiation from the atmosphere and Earth’s surface, forming sky background radiation. Considering the response wavelength of the single-photon detector system, the background noise photon rate per unit time, $n_{n}$, received by the system can be expressed as [17]:(1)$$n_{n}=\frac{\pi \cdot \lambda }{4c\cdot h}I_{SR}\cdot \theta_{r}^{2}\cdot A_{r}\cdot \delta_{\lambda }\cdot K_{r}\cdot \eta_{q}+n_{laser}$$
where


$I_{SR}$: Background brightness,$\theta_{r}$: Receiving field-of-view angle,$\delta_{\lambda }$: Bandwidth of the narrowband filter,λ: Wavelength,h: Planck’s constant,c: Speed of light in vacuum,$A_{r}$: Effective area of the receiving mirror,$K_{r}$: Efficiency of the receiving optical system,$\eta_{q}$: Echo photon detection efficiency (quantum efficiency),$n_{laser}$: Detector’s intrinsic noise.


In the absence of a signal, the probability of noise occurring within a single detector opening time $\tau_{r}$ can be described by(2)$$P_{noise}=\sum_{k=1}^{n}\frac{e^{-\lambda }\lambda^{k}}{k!}=1-e^{-n_{n}\tau_{r}}$$

According to the properties of the Poisson process, the probability density of an event occurring at time t represents the first occurrence time of the event. Taking the detector activation moment as the initial time (t=0), the distribution function $f_{noise\left(t\right)}$ for the time interval between the noise detection and the detector activation can be expressed as follows:(3)$$f_{noise}\left(t\right)=dtdP\left(T\leq t\right)=n_{n}e^{{-n}_{n}t}$$

### 2.2. Echo Distribution Model

In satellite laser ranging (SLR), the photon echo phenomenon exhibits a shape effect that can degrade the ranging accuracy [17,18]. However, in space debris laser ranging applications, this shape-induced distortion becomes negligible due to the specific operational parameters. Typically employing high-power pulsed lasers with single-pulse durations around 10 ns, such systems achieve a theoretical spatial resolution of approximately 1.5 m (calculated as c×τ/2, where c is the light speed and τ is the pulse duration). This resolution threshold corresponds to or slightly exceeds the characteristic dimensions of most tracked debris targets (>1 m in size).

The fundamental mitigation mechanism lies in the comparative scaling between the pulse characteristics and target geometry. For typical debris objects exceeding 1 m in size, the shape-induced pulse broadening (estimated at 1–2 ns) becomes proportionally insignificant relative to the primary ranging interval. This temporal distortion corresponds to spatial uncertainties of 15–30 cm (equivalent to 0.15–0.3 m through the c×τ/2 conversion), which represent less than 20% of the inherent system resolution. Moreover, the extended pulse duration inherently reduces the susceptibility to waveform distortion compared to the sub-nanosecond pulses used in traditional SLR for cooperative satellites with retroreflectors.

This operational paradigm shift is enabled by two key factors: (1) The non-cooperative nature of the debris targets necessitates higher pulse energies (typically in the 100 mJ to 1 J range) to compensate for the diffuse reflection losses, which naturally aligns with longer pulse durations to maintain safe power densities in the laser systems. (2) The reduced precision requirements for debris tracking (centimeter-level versus millimeter-level for satellite geodesy) permit the tolerance of minor waveform distortions. Experimental validations using orbital debris simulators have demonstrated that the residual shape effects contribute less than 5% to the total ranging error budgets under typical observation geometries.

In space debris laser ranging systems, the interaction between the incident laser pulses and target surfaces follows complex scattering dynamics. When the laser irradiates the orbital debris, the return signal reaches the telescope receiver primarily through diffuse reflection mechanisms. This is particularly true for the majority of tracked debris objects—including spent rocket upper stages and defunct satellites—which exhibit surface characteristics dominated by either intrinsic roughness or acquired degradation from prolonged space exposure. Key degradation factors include atomic oxygen erosion, ultraviolet-induced polymer embrittlement, and micrometeoroid impact pitting, all of which enhance the diffuse scattering properties.

Consequently, space debris can be appropriately characterized as Lambertian scatterers obeying the cosine radiation law. The mean photoelectron count generated per laser pulse during space debris ranging is given by [17]:(4)$$n_{e}=\frac{n_{0}}{l_{laser}}=\frac{\lambda_{laser}\eta_{q}}{hc}\cdot \frac{E_{l}A_{r}\sigma \rho cos\beta }{4\pi \theta_{t}^{2}R^{4}l_{laser}}\cdot T^{2}\cdot K_{t}\cdot K_{r}$$
where


$n_{0}$: Average number of photoelectrons per laser pulse,$l_{laser}$: Laser pulse width,$\lambda_{laser}\eta_{q}$r: Wavelength of the emitted laser,$E_{l}$: Laser power,σ: Reflective cross-sectional area of the target,ρ: Target reflectivity,cosβ: Lambertian cosine factor (β is the angle between the Lambertian radiation direction and the target surface normal),$\theta_{t}$: Laser beam divergence angle,R: Radial distance to the target,T: Single-pass atmospheric transmittance,$K_{t}$: Efficiency of the transmitting system,$K_{r}$: Efficiency of the receiving system.


In the absence of noise, the probability of echo photons successfully reaching the detector can be expressed as follows:(5)$$P_{echo}=\sum_{k=1}^{n}\frac{e^{-\lambda }\lambda^{k}}{k!}=1-e^{-n_{e}\tau_{r}}$$

The distribution function for the arrival time t of the echo photons relative to the detector activation moment is(6)$$f_{echo}\left(t\right)=n_{e}e^{-n_{e}t}$$

### 2.3. Measurement Data Distribution Model

According to the operational principles of single-photon detectors, the detector responds identically to noise and valid echo signals. Specifically, after each range gate activation, the detector only registers the first arriving photon, whether from noise or an echo. Consequently, during actual observations, the measured data may contain both noise and echo signals. The background noise acts as a continuous signal persisting throughout the detector’s response time. Since the noise and echo photons are statistically independent events, and neglecting the shape effects of the observed target, the temporal distribution of the echo signal matches the width of the emitted laser pulse. Assuming the echo photon return time relative to the detector activation moment is $t_{f}$, the distribution function of the measurement data can be modeled as a joint distribution of the noise and echo photons based on the Poisson statistics:(7)$$\left\{\begin{array}{cc} f_{meas}\left(t\right)=n_{n}e^{{-n}_{n}t} & t<t_{f}\\ f_{meas}\left(t\right)=\left(1-e^{{-n}_{n}t_{f}}\right){(n}_{e}{{+n}_{n})e}^{-{(n}_{e}{+n}_{n})t} & t_{f}\leq t\leq t_{f}+l_{laser}\\ f_{meas}\left(t\right)=\left(1-e^{{-n}_{n}t_{f}}\right)\left(1-e^{-{(n}_{e}{+n}_{n})(t_{s}-t_{f})}\;\;\right)n_{n}e^{{-n}_{n}t} & t_{f}+l_{laser}<t\;\; \end{array}\right.$$

### 2.4. Analysis of Measurement Data Distribution Model

In practical measurements, the observed data consist of discrete points from multiple measurements. To perform a distribution analysis, the data can be partitioned into intervals as follows:


1.Convert the observed O-C data into the relative time data ∆t, referenced to the range gate activation moment (t=0).2.Divide the time axis into intervals of the length $l_{i}$. Assuming negligible pulse broadening due to the target shape effects, the echo duration matches the laser pulse width $l_{laser}$. When $l_{i}<l_{laser}$, the echo signals may disperse across multiple intervals, leading to the failure of distribution deviation detection. Conversely, when $l_{i}\gg l_{laser}$, the signals concentrate in a single interval, but the geometric properties of the noise distribution may be contaminated by the signals within the interval. Experimental results show that the optimal $l_{i}$ should satisfy $l_{i}=\alpha \cdot l_{laser}(\alpha \in [1.2,2.0])$, balancing the signal concentration and noise distribution stability. To ensure the echo falls within one or two intervals, set $l_{i}\geq l_{laser}$.3.Define the detector’s operational time window as $\left[0,\tau_{r}\right]$. This window is partitioned into $\lfloor \frac{T}{l_{i}}\lfloor$ intervals: $\left[0,l_{i}\right],\left[l_{i},2l_{i}\right],\cdots \left[\left(N-1\right)l_{i},Nl_{i}\right]$, yielding the dataset $T=\left\{∆t_{1},∆t_{2}\ldots \ldots ∆t_{N}\right\}$.


If the dataset contains no echo signals, the probability $p_{D}\left(i\right)$ of the first noise event occurring in the ii-th interval $\left[il_{i},\left(i+1\right)l_{i}\right]\left(i=0,1,2,\cdots \right)$ is derived from Equation (3).

The ratio of probabilities between adjacent intervals is constant:(8)$$\frac{p_{D}\left(i+1\right)}{p_{D}\left(i\right)}=\frac{e^{-n_{n}\left(i+1\right)l_{i}}\left(1-e^{-n_{n}l_{i}}\right)}{e^{-n_{n}il_{i}}\left(1-e^{-n_{n}l_{i}}\right)}=e^{-n_{n}l_{i}}$$

This implies that the interval counts follow a geometric distribution with the probability mass function below:(9)$$P\left(X=k\right)={\left(1-p\right)}^{k-1}p$$
where(10)$$p=1-e^{-n_{n}l_{i}}$$(11)$${\left(1-p\right)}^{k-1}=e^{-n_{n}\left(k-1\right)l_{i}}$$

When the measurement data include echo signals (assumed to occur in the k-th interval $\left[kli,\left(k+1\right)li\right])$, the data distribution model must account for the combined effects of the noise and echoes. The distribution is partitioned into three phases:


Pre-Echo Intervals i<k


These intervals are influenced solely by the noise. The detection probability aligns with the noise-only case:(12)$$p\left(i\right)=e^{-n_{n}il_{i}}\left(1-e^{-n_{n}l_{i}}\right)$$

The ratio of probabilities between adjacent intervals remains constant:(13)$$\frac{p\left(i+1\right)}{p\left(i\right)}=e^{-n_{n}l_{i}}$$


Echo-Active Interval i=k


Within this interval, both the noise and echo photons contribute, resulting in a combined photon arrival rate of $n_{e}{+n}_{n}$. The detection probability becomes(14)$$p\left(k\right)=e^{-n_{n}kl_{i}}\left(1-e^{-{(n}_{e}{+n}_{n})l_{i}}\right)$$

The probability ratio relative to the previous interval is as follows:(15)$$\frac{p\left(k\right)}{p\left(k-1\right)}=e^{-n_{n}kl_{i}}\frac{1-e^{-{(n}_{e}{+n}_{n})l_{i}}}{\left(1-e^{-n_{n}l_{i}}\right)}$$

Key Observation: This ratio is no longer constant, violating the geometric distribution property.


Post-Echo Intervals i>k


If the detector is not triggered in interval k (with a probability of $e^{-{(n}_{e}{+n}_{n})l_{i}}$), the subsequent intervals revert to the noise-only distribution, but must account for the cumulative probability of no prior triggering:(16)$$p\left(i\right)=e^{-n_{n}kl_{i}}\cdot e^{-{(n}_{e}{+n}_{n})l_{i}}\cdot e^{-n_{n}\left(i-k-1\right)l_{i}}\left(1-e^{-n_{n}li}\right)$$

The simplified form is as follows:(17)$$p\left(i\right)=e^{-n_{e}li}\cdot e^{-n_{n}\left(i-1\right)li}\left(1-e^{-n_{n}li}\right)$$

The probability ratio between adjacent intervals recovers to a constant:(18)$$K=\frac{p\left(i+1\right)}{p\left(i\right)}=e^{-n_{n}l_{i}}$$

#### Practical Implications

In actual measurements, due to the uncertainties of space debris orbit prediction, the occurrence of echo signals at the initial stage is typically a low-probability event. Thus, the early-stage data predominantly follow the noise distribution. The key conclusions from the above derivations include the following:


After partitioning the data into intervals, the ratio $\frac{p_{D}\left(i+1\right)}{p_{D}\left(i\right)}$ remains constant at $e^{-n_{n}l_{i}}$ under noise-only conditions.At a specific adjusted range deviation time of $T_{R}$, the data distribution pattern deviates due to the echo interference.


However, during measurement, the angle between the target and the Sun continuously changes, causing dynamic variations in the background noise intensity. Additionally, multiple factors influence the noise intensity, making accurate predictions challenging. Therefore, the proposed methodology involves the following:


Distribution Fitting: Use the measurement data to empirically fit the noise distribution.Deviation Detection: Identify the intervals where the data significantly deviate from the fitted distribution. These intervals (typically one or two) indicate the presence of echo signals.Back-to-O-C Conversion: Map the identified time regions back to the O-C data to obtain the coarse measurements.


## 3. Algorithm Design

The key to designing an adaptive identification algorithm for daytime space debris laser ranging data lies in evaluating the data distribution characteristics. The parameters of the distribution function can be estimated via maximum likelihood estimation (MLE). Let $n_{k}$ denote the number of data points W falling in the k-th interval and K represent the total number of intervals. Based on Equation (9), the likelihood function is as follows:(19)$$L\left(p\right)=\prod_{k=1}^{K}{\left[{\left(1-p\right)}^{k-1}p\right]}^{n_{k}}$$

Taking the natural logarithm of the likelihood function,(20)$$\ln L\left(p\right)=\sum_{k=1}^{K}n_{k}\left[\ln p+\left(k-1\right)\ln\left(1-p\right)\right]$$

To find the MLE estimate $\hat{p}$, take the derivative of $\ln L\left(p\right)$ with respect to p and set it to zero:(21)$$\hat{p}:\frac{d\ln L\left(p\right)}{dp}=\frac{1}{p}\sum_{k=1}^{K}n_{k}-\frac{1}{1-p}\sum_{k=1}^{K}\left(k-1\right)n_{k}=0$$

Solving Equation (21) yields $\hat{p}$^. Using$\;\hat{p}$, compute the expected count $E_{j}$ for each interval group:(22)$$E_{j}=N\cdot \left[F_{X}\left(k_{max}^{j}\right)-F_{X}\left(k_{min}^{j}-1\right)\right]$$
where $F_{X}\left(k\right)=1-{\left(1-\hat{p}\right)}^{k}$ is the cumulative distribution function (CDF) of the geometric distribution.

The geometric distribution’s interval proportionality coefficient is then(23)$$K=e^{-n_{n}l_{i}}=\frac{E_{j+1}}{E_{j}}$$
with the degrees of freedom being as follows:(24)df=r−1−1=r−2

### 3.1. Chi-Square Test and Echo Identification


Critical Value Determination:


Obtain the critical value $\chi_{\alpha ,df}^{2}$ from the chi-square distribution table at a given significance level α.


Confidence Interval via Normal Approximation:


Calculate the confidence interval for the observed frequencies using the normal approximation method. If the chi-square statistic satisfies $\chi^{2}<{(\chi }_{\alpha ,df}^{2}+z\sqrt{\chi_{\alpha ,df}^{2}})\;$(where z is the standard normal quantile) within an acceptable error margin, the data can be considered consistent with the hypothesized geometric distribution (noise-only scenario).


1.Deviation Detection:


If the data deviate from the geometric distribution, identify the interval with the maximum contribution to the chi-square statistic. This interval exhibits the largest discrepancy between the observed $O_{j}$ and expected $E_{j}$ frequencies, indicating potential echo signals.


2.O-C Conversion:


Map the identified interval to its corresponding time relative to the range gate activation. This yields the coarse reception time of the echo signal, enabling real-time feedback of the measurement data from the preceding second.

### 3.2. Operational Workflow


1.Data Acquisition:


Begin recording the measurement data once the target reaches a suitable observational angle.


2.Distribution Testing:


Perform distribution testing on the most recent 1 s data window. Estimate the noise intensity $n_{n}$ via the proportionality coefficient $K=e^{-n_{n}l_{i}}$, which also quantifies the observational success probability.


3.Decision Logic:Consistent with Noise: If the chi-square test validates the geometric distribution, proceed with the noise-only parameter updates.Echo Detected: If the deviations are significant,Extract the interval with the highest chi-square contribution.Compute the corresponding O-C values and output results.Validate the results by checking the consistency over consecutive time windows.4.Real-Time Output:


Persistently clustered outputs within a defined timeframe are flagged as valid measurements, achieving the real-time identification of space debris laser ranging in daylight conditions.

## 4. Verification and Result Analysis

### 4.1. Noise Model Verification

To validate the effectiveness of the quasi-real-time identification algorithm for daytime space debris laser ranging data, we first conducted verification using the daytime laser ranging noise data. The experimental data were acquired from daytime ranging trials on the space debris. The system employed a high-power pulsed laser with an 8 ns pulse width and an operating frequency of 400 Hz. The target observation time windows and the angular separation ranges between the targets and the Sun are summarized in Table 1.

For each target, four sets of eight-second continuous data segments from different time windows were selected and processed using the proposed real-time identification algorithm. The corresponding results are illustrated in Figure 1 and Figure 2:

The noise intensities for all three targets fluctuated around 2 × 10^7^ s^−1^, with a noticeable decrease observed when the angular separation from the Sun was larger. This confirms that the noise intensity derived from the algorithm aligns with the real-world conditions.

As shown in the figures, while distribution anomalies were detected in certain data segments, the discontinuous intervals of the identified results across consecutive computations suggest that the outputs are attributable to random measurement errors rather than the presence of echo signals.

### 4.2. Simulated Echo Validation

Building on the verification that the noise in the measurements conforms to the distribution model, we further validated the algorithm’s capability to detect echo signals using Monte Carlo simulations.


1.Equal Echo and Noise IntensitiesSimulated data were generated with $\frac{n_{e}}{n_{n}}=1$ (i.e., 0.16 echo photons per pulse) under varying laser pulse intensities.As shown in Figure 3, the algorithm’s identification performance degrades as the range gate duration increases. This occurs because the probability of the echo signals falling within a single interval diminishes, reducing their impact on the overall distribution.



2.Varying Echo-to-Noise RatiosMonte Carlo simulations were conducted for $\frac{n_{e}}{n_{n}}=1\sim 0.2$.Figure 4 demonstrates that the algorithm reliably identifies echo signals whenThe echo photon intensity exceeds 60% of the noise intensity ($n_{e}$ > 0.09 photons/pulsene),The local noise density surpasses the echo density.


### 4.3. Data Segmentation and Analysis


1.Data SelectionA segment of the measurement data containing both manually identified echo regions and noise-only regions was extracted (Figure 5).The vertical axis represents the O-C values, with the red horizontal lines marking the range gate activation times.



2.Manual ValidationA post hoc manual analysis confirmed valid echo signals between 07:56:18 and 07:56:25, during which the system continued adjusting the range gate due to the lack of real-time feedback.



3.Algorithm Testing:Noise-Only Data: Four 4 s noise segments were processed.Echo-Containing Data: Four groups of echo data (post-range gate adjustment) were analyzed.4.Results


Figure 6 and Figure 7 show the algorithm’s outputs.

Noise Segments: The algorithm correctly identified the noise-only distributions.

Echo Segments: Echo signals were detected with high confidence, aligning with the manual validation results.

The quasi-real-time processing capability met the requirements for real-time feedback in daytime ranging operations.

### 4.4. Comparative Analysis of Methods

Dependence on Geometric Distribution Assumption and Impact of Non-Poisson Noise: The proposed algorithm fundamentally relies on the assumption that the background noise follows a geometric distribution, derived from the Poisson process governing photon arrivals. However, real-world deviations from the Poisson statistics—such as detector dead-time effects—introduce critical challenges.

Dead-Time Effects: During the detector’s dead-time period (typically nanoseconds to microseconds after each photon detection), the subsequent photons are either ignored or recorded with reduced efficiency. This violates the Poisson independence assumption, leading to sub-Poissonian statistics where the variance of the photon counts is suppressed. Consequently, the geometric distribution model for the noise intervals (Equations (8)–(11)) becomes inaccurate, causing the overestimation of p (detection probability) and false identification of noise as signals.

To systematically evaluate the performance boundaries and applicability of the proposed algorithm, Table 2 compares the statistical distribution method, traditional threshold-based method, and machine learning approaches across key metrics, including core principles, advantages, limitations, and experimental performance. The results demonstrate the following:


1.Comparison with Traditional Threshold-Based MethodsFailure in High-Noise Scenarios: At a background noise rate of $n=2\times 10^{7}$ photons/s, the traditional threshold-based methods fail completely (false detection rate > 90%) due to the signal intensity ($n_{e}=0.09$ photons/pulse) being overwhelmed by the noise. In contrast, the proposed algorithm successfully identifies weak signals by analyzing the statistical distribution deviations.Adaptability to Dynamic Noise: Threshold methods rely on manual parameter tuning and cannot adapt to dynamically changing noise (e.g., fluctuations caused by varying Sun–target angles). The proposed method, however, achieves adaptive recognition through real-time distribution fitting and chi-square testing.2.Complementary Potential with Machine LearningGeneralization vs. Data Scarcity: Machine learning (e.g., deep learning) exhibits superior generalization for complex noise patterns, but requires large labeled datasets (currently scarce for daytime ranging) and high computational resources (e.g., GPU acceleration).Lightweight Advantage: The proposed algorithm demonstrates efficiency in data-scarce scenarios (processing within 1 s), making it suitable for real-time applications. Future integration with lightweight online learning frameworks (e.g., incremental neural networks) could further enhance robustness.3.Experimental ValidationDetection Limits: The statistical distribution method achieves the reliable detection of $n_{e}=0.09$ photons/pulse under $n=2\times 10^{7}$ photons/s, whereas threshold-based methods fail entirely in such conditions.Real-Time Feasibility: With a processing time of ≤1 s at 400 Hz, the proposed method outperforms machine learning approaches in latency-critical applications.4.Key TakeawaysPerformance Boundaries: The statistical distribution method breaks the detection bottleneck in high-noise regimes ($n_{n}>n_{e}$), which traditional methods cannot address.Practical Trade-Offs: While machine learning holds theoretical promise, its reliance on labeled data and computational resources limits its practicality for the current daylight ranging scenarios.Synergy Opportunities: Hybrid frameworks combining statistical models with lightweight machine learning could balance adaptability and real-time performance.


## 5. Conclusions

To address the challenge of echo signals drowned in strong background noise during daytime space debris laser ranging, this paper proposes a real-time identification algorithm based on a data distribution model. Through theoretical derivation and experimental validation, the following conclusions are drawn:


1.Joint Noise–Echo Distribution Model


A statistical model integrating single-photon detection characteristics was established, revealing that


The noise data strictly follow a geometric distribution,The hybrid noise–echo data deviate from geometric regularity due to the echo interference.


This theoretical foundation enables robust signal–noise differentiation.


2.Algorithm Performance


The proposed interval partitioning and chi-square goodness-of-fit method achieves


Real-time echo interval identification within 1 s data windows at a laser frequency of 400 Hz,Reliable detection under echo-to-noise intensity ratios as low as 0.6 (minimum ne > 0.09 photons/pulse),Coarse ranging outputs with errors < 3 m (10 ns), meeting real-time operational requirements.



3.Methodological Advancement


Unlike the conventional intensity-based approaches, this algorithm exploits statistical distribution disparities, overcoming the limitation of noise density surpassing signal density. It provides a novel paradigm for weak signal extraction in photon-starved regimes.

This research advances autonomous space debris tracking, high-precision orbit prediction, and spacecraft safety, with broad implications for space situational awareness.

## Figures and Tables

**Figure 1 sensors-25-02281-f001:**
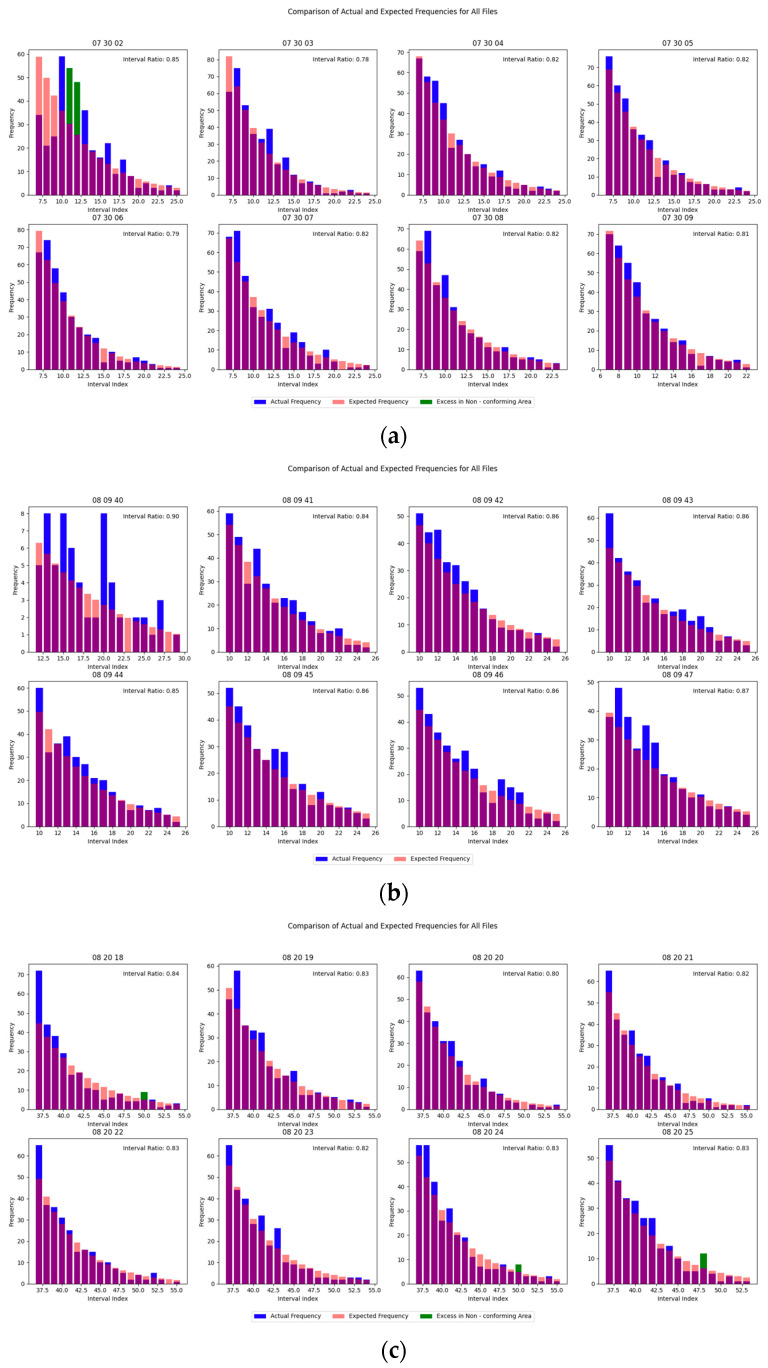
Validation results of data distribution. (**a**) 31793; (**b**) 16720; (**c**) 23705.

**Figure 2 sensors-25-02281-f002:**
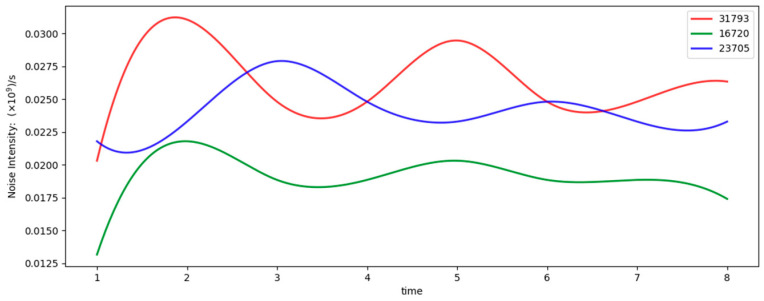
Displays the noise intensity calculated by the algorithm.

**Figure 3 sensors-25-02281-f003:**
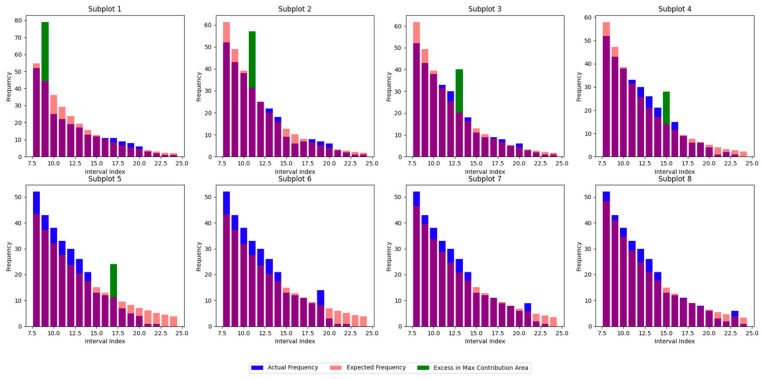
Echo signals appear in different intervals.

**Figure 4 sensors-25-02281-f004:**
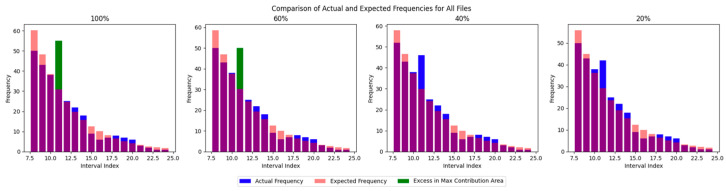
Echo signals with different intensities.

**Figure 5 sensors-25-02281-f005:**
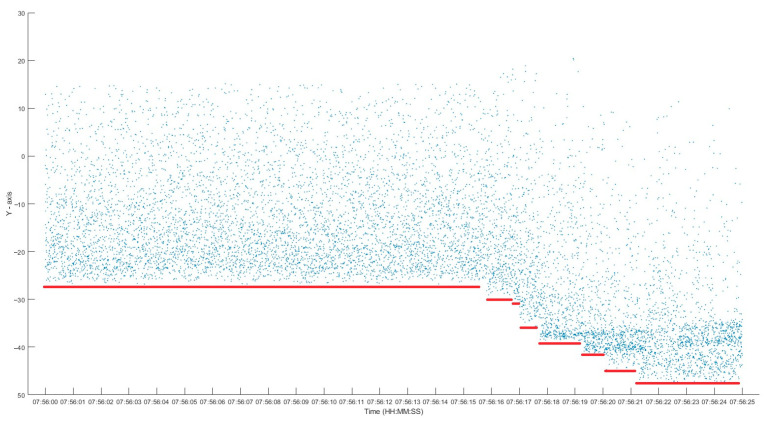
O-C values of observation data.

**Figure 6 sensors-25-02281-f006:**
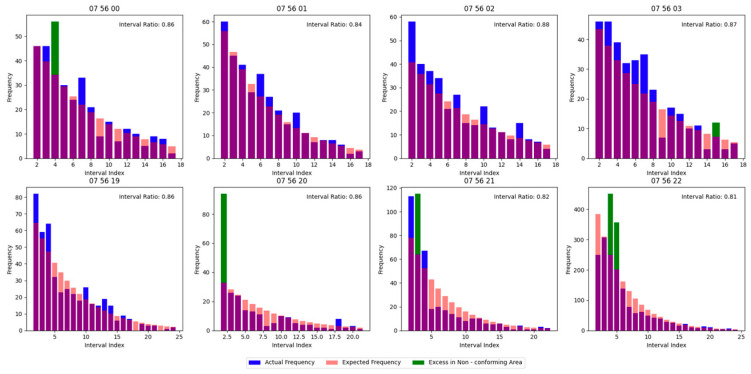
Validation results of data distribution.

**Figure 7 sensors-25-02281-f007:**
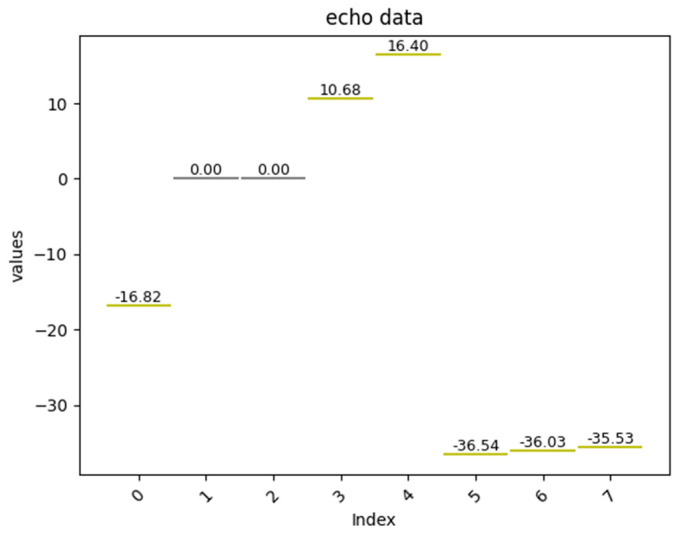
Output result interface.

**Table 1 sensors-25-02281-t001:** Target–sun angular separation ranges.

Nord ID	Time (UTC)	Ranges (°)
31793	07:30:02–07:30:09	93.83–94.74
16720	08:09:40–08:09:47	105.57–107.38
23705	08:20:18–08:20:25	68.18–68.37

**Table 2 sensors-25-02281-t002:** Method comparison: statistical distribution method vs. traditional threshold-based method vs. machine learning method.

	Statistical Distribution Method (Proposed)	Traditional Threshold-Based Method	Machine Learning Method (e.g., Deep Learning)
Core Principles	Detects anomalies via statistical distribution differences (geometric vs. hybrid distribution) using chi-square tests.	Sets fixed intensity thresholds; signals above thresholds are classified as valid.	Learns complex noise–signal patterns through data-driven models for classification/regression.
Advantages	1. No reliance on signal intensity thresholds; works when noise density > signal density. 2. Real-time (≤1 s processing). 3. No training data required.	1. Simple implementation. 2. Effective in stable, high-SNR environments (e.g., nighttime).	1. Captures nonlinear noise patterns. 2. Adapts to dynamic environments (e.g., nonlinear noise variations).
Limitations	1. Assumes noise follows geometric/Poisson distribution. 2. Sensitive to model mismatch (e.g., nonstationary noise).	1. Fails when noise density > signal density. 2. Manual threshold tuning; poor generalization.	1. Requires large labeled datasets (scarce for daytime ranging). 2. High computational costs.
Applicable Scenarios	Low SNR, dynamic noise (e.g., daylight space debris tracking).	High SNR, stable noise (e.g., nighttime satellite ranging).	Complex noise patterns with sufficient labeled data.
Data Requirements	No training data; relies on real-time observations.	No training data; requires empirical threshold tuning.	Large labeled datasets (noise–signal spatiotemporal labels).
Computational Complexity	Low (suitable for embedded systems).	Extremely low (threshold comparison).	High (GPU acceleration needed for training/inference).
Future Directions	Adaptive windowing and online parameter updates for dynamic noise robustness.	Dynamic threshold adjustment (e.g., adaptive filtering).	Lightweight models (e.g., online learning), synthetic data augmentation, multimodal fusion.

## Data Availability

The datasets generated during this study are not publicly available due to confidentiality agreements with collaborative institutions governing the non-public experimental trials. However, data may be shared upon reasonable request from qualified researchers, subject to written approval by all participating entities.

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
