# Peer review of "Real-Time Identification Algorithm of Daylight Space Debris Laser Ranging Data Based on Observation Data Distribution Model"

_sensors, 2025, doi:10.3390/s25072281_

Round 1
Reviewer 1 Report
Comments and Suggestions for Authors
This paper addresses a critical issue in space situational awareness: the dramatic increase in space debris poses significant threats to in-orbit spacecraft. Satellite Laser Ranging (SLR) technology, known for its high single-point precision, has been extensively utilized for the precise measurement and orbit prediction of space debris. However, traditional SLR methods face limitations due to high background noise during the daytime, restricting observations to mornings, evenings, and nights. This limitation reduces the observation window, making it imperative to develop algorithms that can recognize weak echo signals in real-time amidst high noise environments during the day. This method not only improves the success rate of observation, but also provides a new idea for solving the signal recognition problem in complex noise environments. The paper's theoretical derivation is rigorous and experimental verification is sufficient, which has high academic value and practical application potential. The public release of this research will undoubtedly support spacecraft safety and space debris monitoring efforts.
There are 2 points in the paper that could be further improved:
1、The constantly changing angle between the target and the sun causes the background noise intensity to change dynamically during the measurement process, and although the paper proposes to meet this challenge by distribution fitting and bias detection, the effectiveness and stability of this method still need to be further verified.
2、The advantage of this algorithm over traditional methods is mentioned in the literature in that it utilizes statistical distribution differences rather than signal strength thresholds for identification, but the lack of direct comparisons with other existing techniques (e.g., machine-learning based methods) makes it difficult to comprehensively evaluate its strengths and weaknesses in specific application scenarios.
Reviewer 2 Report
Comments and Suggestions for Authors
Review
Overall, this paper reports a real-time identification algorithm for space debris based on the statistical differences between echo signals and those from the daylight background, making it suitable for publication in this journal. For better readability and clarity, several comments and recommendations are provided below.
1. The paper successfully demonstrates the identification of some space debris through both measurements and simulations. However, it should also provide quantitative results such as detection accuracy, false alarm rate, and processing time.
2. The paper neglects background flux variations as well as potential changes in echo signals that may result from the angular spin of the space debris. Please provide a justification for this omission.
3. The paper is somewhat difficult to follow due to numerous formatting errors in mathematical symbols and equations. Additionally, the frequent use of similar or confusing symbols further hinders readability. Some specific examples are listed below:
1) Line 37, 2x107 2x107
2) Line 117, fnoise(t) -> fnoise(t)
3) Line 131, ββ β
4) Line 140, tt -> t
5) Line 186, kk -> k
6) Line 186, 𝑒−(𝑛𝑛+𝑛𝑒)𝑙𝑖 𝑒−(𝑛𝑛+𝑛𝑒)𝑙𝑖
7) Line 248, 𝐾 = 𝑒−𝑛𝑛𝑙𝑖 𝐾 𝐾 = 𝑒−𝑛𝑛𝑙𝑖
4. Lines 120–123:
The paper assumes that space debris exhibits Lambertian surface emission properties. While this is a common simplification in optical measurements to facilitate calculations and modeling, it may not always be valid, particularly for artificial space objects. Many such objects have surface finishes that deviate from Lambertian behavior. Non-Lambertian surfaces often exhibit specular or directional reflectance characteristics, which can significantly affect the intensity of reflected light received by the telescope. This, in turn, may impact the accuracy of range and reflectance-based measurements. The authors are encouraged to discuss the validity and limitations of this assumption.
5. Line 149:
In addition to the comment above, the paper also overlooks the influence of the target’s shape. The authors are encouraged to clarify whether this simplification is reasonable and justifiable.
6. Lines 243–244 (Data Acquisition section):
The paper states, “Begin recording measurement data once the target reaches a suitable observational angle.” For the proposed method, what specifically constitutes a “suitable observational angle”? Please clarify.
